# Local-Basis-Function Equation of State for Ice VII–X to 450 GPa at 300 K

**J. Michael Brown * and Baptiste Journaux** 

Earth and Space Sciences, University of Washington, Seattle, WA 98195, USA; baptiste.journaux@gmail.com
* Correspondence: brown@ess.washington.edu

**Abstract:** Helmholtz energy of ice VII–X is determined in a pressure regime extending to 450 GPa at 300 K using local-basis-functions in the form of b-splines. The new representation for the equation of state is embedded in a physics-based inverse theory framework of parameter estimation. Selected pressures as a function of volume from 14 prior experimental studies and two theoretical studies constrain the behavior of Helmholtz energy. Separately measured bulk moduli, not used to construct the representation, are accurately replicated below about 20 GPa and above 60 GPa. In the intermediate range of pressure, the experimentally determined moduli are larger and have greater scatter than values predicted using the Helmholtz representation. Although systematic error in the determination of elastic moduli is possible and likely, the alternative hypothesis is a slow relaxation time associated with changes in proton mobility or the ice VII to X transition. A correlation is observed between anomalies in the pressure derivative of the predicted bulk modulus and previously suggested higher-order phase transitions. Improved determinations of elastic properties at high pressure would allow refinement of the current equation of state. More generally, the current method of data assimilation is broadly applicable to other materials in high-pressure studies and for investigations of planetary interiors.

**Keywords:** equation of state; Helmholtz energy; phase transition; ice VII; ice X; NaCl; exoplanets; icy/ocean worlds; local-basis-function; b-spline; Tikhonov inverse

## 1. Introduction

From icy worlds of our solar system to water-rich super-Earth exoplanets, high-pressure water ices are potential planetary constituents that can exist as distinct mineral species. For example, on Earth, ice VII as a mineral inclusion has been identified in a mantle-derived diamond [1]. Having an accurate equation of state representations for all ice phases is a prerequisite in evaluating questions of origins, evolution, and the potential habitability of ocean worlds [2–9]. A particularly challenging domain for equation of state analysis of ices lies between 2.2 GPa (the pressure of the first order transition between tetragonal ice VI and cubic ice VII) and 450 GPa where three-fold compression is accommodated without first order transitions. First order transitions have been predicted to occur in a regime beyond 300–400 GPa. The experimental and theoretical evidence for higher order transitions at lower pressures is further discussed below.

To provide an improved representation of ice, a new framework for equations of state analysis is described that relies on the thermodynamic representation of Helmholtz energy. The approach follows naturally from long-standing ideas. Birch [10] identified four principle means to obtain equation of state representations: (1) quantum-mechanical treatment, (2) the Thomas–Fermi electron gas approximation, (3) semi-empirical laws for interactions of atoms and ions, and (4) thermodynamic relationships involving elasticity. The latter, a thermodynamic approach using a series of expansions truncated at a low order of Helmholtz energy with parameters expressed in terms of the bulk modulus

and its first two pressure derivatives, is the widely practiced approach. This approach can, in addition, be related to special cases of semi-empirical laws for atom/ion interactions [11]. Here, results from quantum mechanical and Thomas–Fermi calculations are combined with measurements within the new framework.

The current work is also guided by the Stacey et al. [11] comments that "Since almost any analytic form may be used to represent a finite data set if sufficient arbitrary or adjustable constants are allowed, we may logically regard as most successful those that use fewest such constants to achieve a particular precision of fit. If this still leaves too much choice, we would prefer a theory in which the adjustable constants are determined most directly from observations." The new framework is responsive to the need to keep the size (number of required parameters) of a model small, while adequately representing measurements, theory, and physical insight garnered from examination of data and calculations. An additional argument developed here is that representational results should not depend on the method of parameterization (i.e., the choice of underlying basis functions) and the size of a model should be flexibly adjusted to meet the physics needs of a particular material.

Our primary findings are straightforward. The traditional physics-blind series expansion to low order of Helmholtz energy, while successful in reproducing measurements and/or theory in restricted regimes of compression, eventually fails over larger regimes. Electronic states and the nature of bonding fundamentally change over a large range of compression. Representations, as developed here, preserve the concept of low-order expansions, using subdomains of compression, while allowing for the necessary changes of the crystal potential (as represented by Helmholtz energy) over larger regimes of pressure and volume compression. Within this framework, concerns that dominate conventional equation of state analysis, including the choice of the expansion order and the definition of a strain metric, become less important. The relevant concerns then appropriately shift to a physics-based discussion: the relative weighting of data from varied sources, the quality of results based on theory, as well as the articulation of appropriate physical insight that should be applied to the representations.

In the following sections we (1) review knowledge concerning the behavior of ice beyond 2.2 GPa, (2) discuss the conceptual foundations of conventional equations of state based on series expansions of Helmholtz energy in powers of strain, (3) describe local-basis-functions that are related to, but more flexible than the polynomial global-basis-functions underlying standard parameterizations, (4) test the new framework against data over an extended regime of pressure for a relatively simple ionic material (NaCl), and finally, (5) use the new framework in analysis of high pressure ices. Mathematical and conceptual details of the framework are given in an exposition on basis functions and inverse theory found in Appendix A.

*Phase Behavior of Ice Beyond 2.2 GPa*

Although continuous in volume, the complex variations in observed physical properties in the stability field of ice VII and ice X are currently understood as changes in proton dynamics and distortions of the oxygen lattice during compression. Details of transitions remain debated and observations that support the existence of higher-order transitions are summarized in Table 1 and are further discussed here. *International Union of Crystallography* "short name" designations are used to describe the space group of ice phases in the following discussion.

Cubic ice VII (*Pn-3m*) is stable beyond 2.2 GPa until reaching an indistinct transition above about 60 GPa to ice X. The gradual symmetrization of hydrogens between oxygens leads to ionic ice X (which is also *Pn-3m*). The stability of cubic ice X above 150 GPa is experimentally unconstrained, but density-functional calculations of ground state structures [12,13] suggest an eventual transition to the *Pbcm* structure between 300–400 GPa, then to *Pmc2$_1$* at 930 GPa and *P2$_1$* at 1.3 TPa. Eventually a metallic transition to *C2/m* is predicted at 4.8 TPa. At higher temperatures, ice X transforms into super-ionic ice XVIII above 2000 K [14]. At pressures below 100 GPa, anomalies are reported associated with changes in derivatives or discontinuities in optical spectra, bulk moduli, proton dynamics, and electrical conductivity. The most prominent features include a suggested softening of ice VII volume

around 40 GPa [15–17] and the onset of the ice X transition by proton symmetrization above about 65 GPa. The latter is supported by changes in the IR reflectivity trend of $v_3'$ and translation modes $v_T$ [18], optical reflectivity [19] and from H-NMR experiments [20].

**Table 1.** Reported transitions in ice VII and ice X.

| Pressure of Transition | Suggested Transition | Type of Measurements | References |
|---|---|---|---|
| **5 GPa** | | | |
| 5 GPa | Tetragonal distortion | Powder X-Ray diffraction (XRD) | Grande et al. [21] |
| **10–15 GPa** | | | |
| 11 GPa | Lattice distortion | XRD peak splitting | Hirai et al. [22] |
| 14 GPa | Strain in cubic lattice | Powder X-Ray diffraction | Somayazulu et al. [23] |
| 11 GPa | - | Changes in Raman line width trends for the $v_1(A_{1g})$ band | Hirai et al. [22], Pruzan et al. [24] |
| 13–15 GPa | - | Raman line pressure trends | Zha et al. [25] |
| 13 GPa | - | Neutron diffraction (220/110 ratio) | Guthrie et al. [26] |
| 10–14 GPa | Lattice distortion | c/a ratio changes in ice VIII | Yoshimura et al. [27] |
| 10–14 GPa | - | Maximum in electrical conductivity | Okada et al. [28] |
| 10–15 GPa | - | Maximum in proton diffusion | Noguchi et al. [29] |
| **20–25 GPa—Possible transition to Ice VII′ with proton dynamic disorder (tunneling and thermal hoping)** | | | |
| 23–25 GPa | - | Bump in the 220/110 ratio from Neutron diffraction | Guthrie et al. [26] |
| 25 GPa | Proton tunneling: ice VII′ | IR reflectivity trend of $v_3$ and $v_3'$ trend | Goncharov et al. [18] |
| 20–25 GPa | Proton tunneling: ice VII′ | H-NMR | Meier et al. [20] |
| 27 GPa | - | Raman line pressure trends | Zha et al. [25] |
| **40 GPa** | | | |
| 40 GPa | Softening | Drop in volume reported based on XRD | Hemley et al. [15], Loubeyre et al. [16], Sugimura et al. [17] |
| 44 GPa | - | Raman line pressure trends | Zha et al. [25] |
| 44 GPa | - | Discontinuity in the pressure dependence of Brillouin esound speeds | Noguchi et al. [30] |
| 40 Gpa | - | Changes in trend of reflective index | Zha et al. [19] |
| 40 GPa | - | Drop in Brillouin transverse wave speeds over a narrow P range (<2 GPa) in compression and decompression | Asahara et al. [31] |
| **>60 GPa transition to ice X** | | | |
| 60 GPa | Proton symmetrization | IR reflectivity trend of $v_3'$ and translation modes $v_T$ | Goncharov et al. [18] |
| 62 GPa | - | Raman line pressure trends | Zha et al. [25] |
| 60 Gpa | - | Changes in trend of reflective index | Zha et al. [19] |
| 59 GPa | - | Drop in Brillouin transverse wave speeds in compression | Asahara et al. [31] |
| 70 GPa | Proton symmetrization | H-NMR | Meier et al. [20] |
| 90 GPa | Proton symmetrization | Emergence of the p20 Raman mode | Zha et al. [25] |

Several other potential higher order transitions have been identified. Tetragonal distortion is possible above 5 GPa [21]. Between 10 GPa and 14 GPa shifts are observed in X-Ray, Neutron, and Raman measurements [22–24,26]. Such behavior is also observed in the proton-ordered analog of ice VII, known as ice VIII [27]. The initiation of proton tunneling above 25 GPa (possible transition to ice VII′) is interpreted on the basis of IR reflectometry [18] and H-NMR data [20].

Equations of state for ice VII and X based on density functional theory, developed by French et al. [32], show variable degrees of agreement with measurements at 295 K depending on the chosen potential and the range of density. They noted that "the fitting of experimental data for such a complex solid may require using a more refined parametric equation of state than the simple Vinet formula, even when the equation of state is split into multiple sections [33]".

As a result of the complex behavior noted here, ice VII and X viewed as a single-phase continuum, cannot be appropriately represented by a single (relatively low order) global polynomial-based equation of state. Previous equations of state, constrained by measurements over limited and differing ranges, present a confusing collection of parameterizations. Extrapolations using any of these equations of

states to higher or to lower pressures than the range of fitted data results in significant misfits of other measurements.

## 2. Materials and Methods

### 2.1. Helmholtz Energy-Based Equations of State

Here, a brief overview of conventional (global) equation of state parameterizations is provided to differentiate them from the local-basis-function method described later. The derivation of an equation of state based on a series expansion of Helmholtz energy as a function of a strain metric, $\eta$, has been extensively discussed [10,11,34–36]. The choice of strain metric is arbitrary, and several have found favor in successful applications of data representation, although the finite strain metric based on an Eulerian representation (as opposed to Lagrangian) is commonly adopted. The Eulerian finite strain metric (as used in the Birch–Murnaghan equation of state) is $\eta = 1/2((V/V_o)^{-2/3} - 1)$, where $V$ is volume and subscript refers to the ambient pressure value. The Vinet strain metric [36] is $\eta = ((V/V_o)^{-1/3} - 1)$. Poirier and Tarantola [35] developed a logarithmic equation of state using $\eta = -1/2 \log (V/V_o)$. Other definitions are possible and have been used.

The expansion of the Helmholtz energy about ambient pressure as a function of the strain metric is given as

$$F(\eta) = F_o + F_1 \eta^1 + F_2 \eta^2 + F_3 \eta^3 + F_4 \eta^4 + \dots \tag{1}$$

where the expansion coefficients $F_i$ contain appropriate derivatives of $F$ evaluated at $\eta = 0$. The first term on the right establishes the energy datum. Pressure follows as

$$P(V) = -\frac{dF}{dV} = -\frac{dF}{d\eta}\frac{d\eta}{dV} = \frac{d\eta}{dV}\left(F_1 + 2F_2\,\eta + 3F_3\eta^2 + 4F_4\eta^3 + \dots\right) \tag{2}$$

The $F_1$ term is conventionally set to zero, although more formally Equation (2) should predict ambient pressure (0.1 MPa) for $\eta = 0$. The isothermal bulk modulus given as

$$K = -V\frac{dP}{dV} \tag{3}$$

has pressure derivatives indicated by the number of prime superscripts:

$$K' = \frac{dK}{dP} \quad K'' = \frac{d^2K}{dP^2} \quad K''' = \frac{d^3K}{dP^3}.$$

Based on one definition for strain, the Eulerian strain metric, and with algebraic effort, Equation (2), truncated at the fourth order in energy, can be expressed in terms of the ambient pressure bulk modulus and its pressure derivatives (denoted with subscript) as:

$$P = \frac{9K_o}{16}\left(-B_1\left(\frac{V}{V_o}\right)^{-5/3} + B_2\left(\frac{V}{V_o}\right)^{-7/3} - B_3\left(\frac{V}{V_o}\right)^{-9/3} + B_4\left(\frac{V}{V_o}\right)^{-11/3}\right) \tag{4}$$

$$B_1 = K_o K_o'' + (K_o' - 4)(K_o' - 5) + 59/9,$$

$$B_2 = 3K_o K_o'' + (K_o' - 4)(3K_o' - 13) + 129/9,$$

$$B_3 = 3K_o K_o'' + (K_o' - 4)(3K_o' - 11) + 105/9,$$

$$B_4 = K_o K_o'' + (K_o' - 4)(K_o' - 3) + 39/9.$$

Equations (2) and (4) contain the common physical assumption that a low order series expansion of energy is adequate. The focus in Equation (2) is on model parameters that are measures of derivatives of the energy at equilibrium. In Equation (4), with application of a particular strain metric, parameters

are cast as elastic properties determined at ambient condition. The use of other strain metrics generates different algebraic relationships but preserves the underlying physics of a truncated series expansion.

While expansion of Equation (1) to arbitrary higher order is straightforward, equation of state applications have been limited to no more than fourth-degree polynomials for several reasons. A general belief has been that an appropriate choice of a strain might lead naturally to a convergent series. Higher order derivatives of the bulk modulus are not well constrained by pressure–volume measurements that may not be sufficiently accurate and/or span a sufficiently large range of compression. The low order (second and third) forms of Equation (4) have provided relatively stable extrapolations to high pressure and have proven adequate to represent many measurements. The fourth-order expansion is often marginally stable or completely unstable in extrapolation. Higher order versions fit to measurements are expected to be less stable. The algebraic and arithmetic effort to relate the energy expansion coefficients to derivatives of the bulk modulus increases with increasing order.

The strength of the strain-expansion-based representation of data is that a small model consisting of three or fewer parameters ($F_2$, $F_3$, $F_4$ or $K_0$, $K_0'$, $K_0''$, plus the 1 bar volume) adequately fits many data sets. The weakness lies in the requirement that the underlying energy potential be represented by a physics-blind series expansion arbitrarily truncated to a small order. Only for a sufficiently small interval will a low order polynomial provide a complete (able to replicate all data within uncertainties) representation. Furthermore, high-pressure behavior may be governed by physics that is not represented in the near ambient pressure potential. In the following, a framework is developed that introduces the use of physics-based constraints and allows construction of equations of state using basis functions having greater flexibility.

### 2.2. Local-Basis-Function Representation of Helmholtz Energy

Numerical methods, developed to meet modern computational speed and accuracy requirements as well as for computer-aided graphics applications, are shown in this section to provide an alternative framework for the representation of Helmholtz energy. Although standard quantities (e.g., trigonometric and logarithmic) can be determined numerically, interpolation of previously calculated values in lookup tables is more efficient. Various interpolation algorithms have been developed and are embedded in both modern hardware and software. Such methods can be applied to representation of Helmholtz energy since any empirically based equation of state is created as an accurate interpolating formula capable of matching all measurements and theory.

The linear interpolation of a list of energies at different volumes is the simplistic approach that can work with sufficiently dense tables. However, accurate smaller models are possible using better interpolating functions and mathematical forms can be chosen to aid construction of representations. Here, piece-wise polynomials on intervals (splines) are used to represent Helmholtz energy over wide ranges of compression. Within each interval, representations are numerically analogous to the conventional series expansion-based equations of state. However, having separate polynomials in different intervals gives greater flexibility in meeting the needs of the underlying physics.

In the following, b-splines [37] are used since they have a number of beneficial numerical properties that are further described below. A b-spline representation, in essence a lookup table, consists of two lists, one containing points defining intervals of the independent variable and a second containing the model parameters. Basis functions then provide the means to interpolate the table. A b-spline of order two (spline order is one greater than the degree of underlying polynomials) is equivalent to linear interpolation of model parameters as a function of the independent variable. In this case, the representation has discontinuous derivatives at interval boundaries. With increasing b-spline order (using higher degree polynomials within each interval) the number of continuous derivatives at interval boundaries increases. The size of a b-spline equation of state is larger than that of a conventional representation. In addition to coefficients required for any global polynomial representation, at least one additional model parameter is needed for each added interval. As shown later, only a few intervals are needed to achieve adequate representations and the model size remains small.

Key properties of b-spline basis functions that make their use advantageous for equation of state representations include:

- B-spline basis function values are available in all computer environments as a call to a function/subroutine. Analogous to the use of exponential or trigonometric functions, no custom (user) programing is necessary for use of b-spline basis functions. The evaluation of equation of state properties then uses universal calling functions that are not material specific.
- The calculation of values and derivatives of a b-spline model are based on linear programing. Interpolation using b-splines is essentially a weighted average of neighboring model parameters with the basis functions providing the normalized weights. This enables efficient computer algorithms for both construction and evaluation of spline models. Arbitrary precision is possible in representing any functional behavior.
- B-spline basis functions are localized. Unlike global polynomial fits of data, spline model parameters pertain to the behavior of the underlying function in a separate restricted regime of the independent variable.
- Details of how intervals are defined allow flexibility in the behavior of function derivatives at interval boundaries. It is possible to allow discontinuities of the function or specified derivatives of the function at a location to meet the needs of a particular equation of state that might involve higher-order transitions.

Additional b-spline details are important in understanding their use for equations of state representations. The full articulation of the underlying mathematics is provided by de Boor [29]. Only a short overview that emphasizes points relevant to the current applications is provided in Appendix A. These include further articulation of spline order, model parameters, and b-spline basis functions. Also included is a description of the interval boundaries, the knots, and how local derivatives are controlled by knot multiplicity. An example representation of Helmholtz energy is given to further illustrate the underlying concepts.

Although spline representations of data can be constructed using standard numerical packages, the concept of local-basis-function (LBF) equation of state representations, as developed here, emphasizes the use of physics-based constraints during their construction. This was introduced in Brown [38], where Gibbs energy was expanded as a function of pressure and temperature. An equation of state for water based on Gibbs energy LBF representations is given in Bollengier et al. [39] and for high pressure ices in Journaux et al. [40]. Here, Helmholtz energy, with volume as the independent variable, is the focus. The representations are extended far beyond the regime of measurement using theory-based constraints for limiting behavior (i.e., a constraint on the pressure derivative of the bulk modulus at infinite pressure based on the Thomas–Fermi electron gas limit).

*2.3. Determination of Helmholtz Energy by Collocation*

Helmholtz energy, a surface in volume and temperature, can be reconstructed through collocation. Collocation is a standard approach for the numerical solution of differential equations in which basis functions are chosen that match differential properties at specified points, the collocation points. In addition, integrating constants must be applied. Here, considering only the volume axis of energy at 300 K, measurements, theory, and physical insight constrain derivatives of the energy surface at specified collocation sites. A sufficiently flexible set of b-spline basis functions then allows representation of any plausible physical behavior for Helmholtz energy and its derivative properties. Here, the arbitrary energy "datum" is set to zero at equilibrium. The challenges lie in the details of data selection, articulation of what constitutes adequate fitting as opposed to overfitting of data, and how to constrain behavior in data-poor regimes.

Such challenges are not unique to equation of state studies and are described within the rubric of parameter estimation using inverse methods [41]. Here, the "damped least square method" or Tikhonov inversion is used to determine model parameters for equation of state representations. The

minimization of one or more side constraints (hereafter called regularization) is added to the standard least square solution. A straightforward approach based on physical insight is to minimize a specified derivative of the model. A smooth model would be one with the smallest second derivatives while still adequately fitting data. In the case of equations of state represented by energy potentials, the second derivative of the Helmholtz energy figures into the determination of the bulk modulus (as shown in Equation (3)), a third derivative is needed to determine $K'$ and a fourth derivative is needed to determine $K''$. Since $K'$ is a frequent quantity of interpretation and since it is typically found to be smooth and trending to an asymptotic value at infinite pressure, minimizing the fourth derivative of the Helmholtz energy emerges as a possible side constraint. Linear inverse methods then allow determination of the model parameters, fitting pressure–volume (and/or bulk moduli) data with a side constraint on the derivative of energy. The explicit minimization of the third derivative of the bulk modulus (to make $K'$ smooth or to require a specific value for $K'$) is undertaken with non-linear analysis. Further details of inverse theory are described in Appendix A.

In summary, a framework is described to find Helmholtz energy by collocation of measurements, theory, and physical insight (through use of regularization) over arbitrarily large regimes of compression. Local-basis-functions (LBF) based on piece-wise polynomials in the form of b-splines are used. A linear problem allows fits of pressure–volume and bulk modulus–volume measurements. Fits involving determinations of the bulk modulus as a function of pressure and its pressure derivatives requires iterative and non-linear analysis. The resulting representations for Helmholtz energy have model parameters in units of energy that can be analytically evaluated for equation of state properties including pressure, the bulk modulus, and derivatives of bulk modulus as a function of volume (or density). The numerical function to evaluate properties based on the LBF representation is universal, no material-specific coding is required.

A number of decisions are required in the creation of an LBF representation for Helmholtz energy. An optimal choice of strain metric may reduce model size. The number of intervals for strain (as well as their distribution and bounds) can be varied. The cost of subdivision is that at least one additional model parameter is required for each added interval of representation. The total parameter count grows slowly since boundary conditions (matching values and derivatives of values at boundaries) reduce the required number of unique parameters. The order of a chosen basis function set is arbitrary and might depend on the physics of a specific material. Smaller intervals allow lower order fits. The derivative of the potential used for regularization is an arbitrary choice. Additional non-linear optimization is required if a specific limiting value for the pressure derivative of the bulk modulus is required.

Using the MATLAB numerical environment, a set of tools based on the concepts given here are provided with the Supplementary Materials. They allow creation and evaluation of LBF equations of state. Scripts using these tools are also provided that reproduce the analyses given in the following sections.

## 3. Results

### 3.1. Equations of State for NaCl

#### 3.1.1. Data and Representations

Measured volumes of NaCl provide an internal pressure calibrant in some high-pressure experiments [42]. The widely used pressure scale for NaCl given by Decker (1971) [43] was based on an assumed two-term pair-wise interaction energy potential. Accurate high-pressure data obtained subsequent to Decker's work exhibited systematic deviations from his pressure scale. Brown (1999) [44] reported an energy potential for NaCl that provides a better representation of accurate high-pressure and high-temperature measurements. Here, properties on the zero Kelvin isotherm, tabulated in [44], provide precise quantities to evaluate the current method of representation. The data and an example script illustrating the analysis given in this section are provided in Supplementary Materials. The

fitting is constrained by both pressure–volume and bulk modulus data. The relative weight of these data was adjusted to provide a satisfactory balance between simultaneous fits of pressures and the bulk moduli as a function of volume.

In Figure 1 the zero-Kelvin densities (panel a), bulk modulus (panel b), and its pressure derivative (panel c) as tabulated in [44] are plotted as a function of pressure. Lines show equation of state predictions based on the model parameters given in Table 2. Data misfits are listed in Table 2. Values extend from negative pressures (constrained by high-temperature measurements at 1 bar) to the high-pressure limit of the NaCl B1 structure.

Properties are plotted in Figure 1 to pressures beyond the stability regime of the B1 structure to examine the behavior of each equation of state in extrapolation. Both conventional global fits (Birch-Murnaghan third and fourth order) and a non-standard global fit (ninth-order), all based on Equation (2), are illustrated. Local-basis-function (LBF) representations (using both Eulerian and log strain metrics) are also shown.

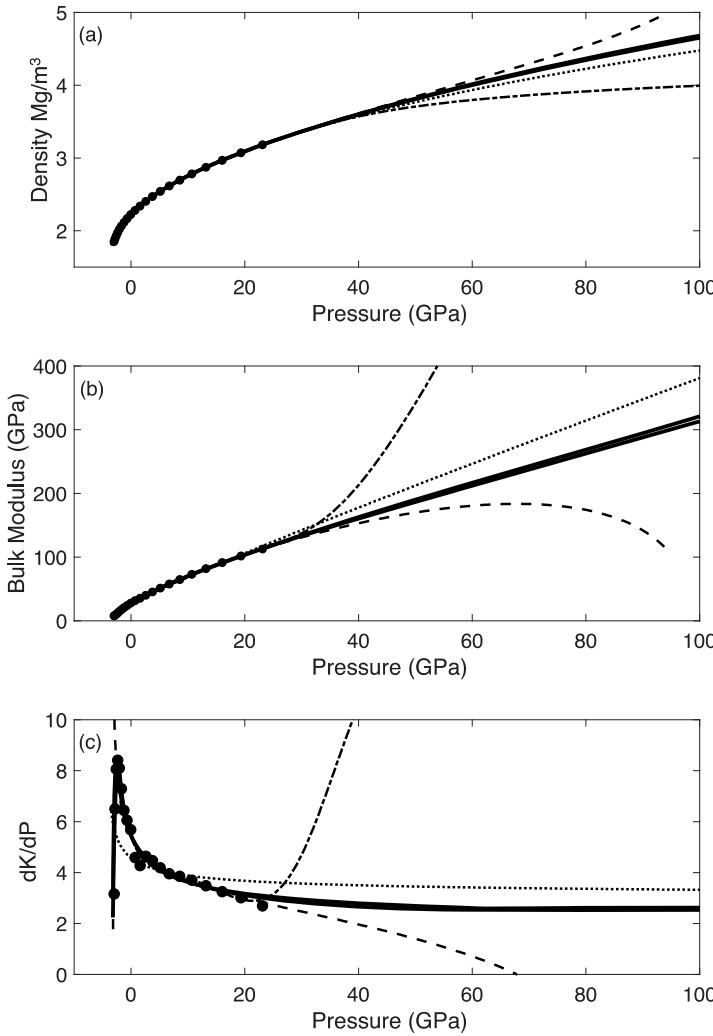

**Figure 1.** Equation of state properties for NaCl as a function of pressure. (**a**) Density. (**b**) Isothermal bulk modulus. (**c**) Pressure derivative of the isothermal bulk modulus. Solid circles in all panels are the tabulated values given in [44]. Lines are predictions based on differing parameterizations as listed in Table 2. Solid lines: local basis functions using different strain metrics (essentially indistinguishable). Dotted line: global third order finite strain. Dashed line: global fourth-order finite strain. Dash-dot line: global ninth-order finite strain.

**Table 2.** Equation of state parameters and root-mean-square (rms) misfits of pressure ($P_{rms}$) and the bulk modulus ($K_{rms}$) for representations of the NaCl 0 K isotherm. A 1 bar density of 2.226 $Mg/m^3$ is used in all representations. Units for all parameters are shown.

| Type of Function | Parameters | | | (rms) |
|---|---|---|---|---|
| **Global-Basis-Function** | | | | |
| Third order/degree Eulerian Finite Strain | $K_o$ = 28.0 GPa | $K_o'$ = 4.5 | | $P_{rms}$ = 0.15 GPa $K_{rms}$ = 2.3 GPa |
| Fourth order/degree Eulerian Finite Strain | $K_o$ = 27.4 GPa | $K_o'$ = 5.4 | $K_o''$ = −0.44 $GPa^{-1}$ | $P_{rms}$ = 0.02 GPa $K_{rms}$ = 0.3 GPa |
| Ninth order/degree Eulerian Finite Strain | $K_o$ = 27.8 GPa | $K_o'$ = 5.4 | $K_o''$ = −0.67 $GPa^{-1}$ | $P_{rms}$ = 0.01 GPa $K_{rms}$ = 0.2 GPa |
| | (Five more parameters for the ninth order fit are not reported here. See Supplementary Materials) | | | |
| **Local-Basis-Function:** | | | | |
| | Knots (strain units): | Coefficients ($GPa\ m^3/Mg$) | | |
| Eulerian Finite Strain Order: 6 (degree 5) | [−0.08, −0.035, 0.24, 0.67] (first and last knots are repeated six times) | [0.308, 0.242, −0.155, −0.292, 4.10, 10.9, 19.8, 26.1] | | $P_{rms}$ = 0.01 GPa $K_{rms}$ = 0.2 GPa |
| log Strain Order: 5 (degree 4) | [−0.09, −0.036, 0.2, 0.42] (first and last knots are repeated five times) | [0.326, 0.241, −0.141, −0.326 3.92 16.4 26.0] | | $P_{rms}$ = 0.01 GPa $K_{rms}$ = 0.2 GPa |

### 3.1.2. Discussion

The energy potential for NaCl over the extended range of volume compression samples significantly different physical behavior. Expanded state properties (negative pressures) are more strongly influenced by long-range cohesive forces (including Coulomb and inductive). In the compressed state, the overlap of electrons creates large short-range repulsive interactions. The large variation of *K'* values with a maximum just below ambient pressure provides evidence that the underlying potential is more complex than one that can be represented by a global low order series expansion.

In Figure 1 all equations of state in the regime of measurements provide adequate approximation to values and trends in density and the bulk modulus. Unsurprisingly, as shown in Table 2, misfit is largest for the third-order representation and this lower order fit has an unsatisfactory representation of the pressure dependence of *K'*. The fourth-order version, while providing a more plausible fit of measurements in the compressed regime, does not have sufficient flexibility to follow the behavior of *K'* in the expanded state. The fourth-order predicted pressure derivative of the bulk modulus continues to increase as pressure becomes increasingly negative. The data show a maximum in *K'* slightly below ambient pressure as required by the necessary behavior of the energy of interaction in the expanded state. As a result of a need for significant curvature of the bulk modulus as a function of pressure near equilibrium (requiring a large second derivative of the bulk modulus), the behavior of the bulk modulus for the global fourth-order fit at high pressure is unstable; *K'* implausibly goes through zero to negative values and the bulk modulus decreases with increasing pressure. A ninth-order fit proved to be the lowest order global representation that could adequately represent all data. As expected, this equation of state cannot be extrapolated.

The LBF equations of state are constructed to have satisfying performance from the expanded state to high compression. They are required to adequately fit data while extrapolating with appropriate physics-based limiting behavior. Two versions (graphically indistinguishable) are shown that fit data equally well. One LBF uses Eulerian finite strains and the other uses log strains. The log strain version allowed use of a lower order spline (order 5) than the Eulerian strains did (order 6). Both LBFs use three intervals (two in the low-pressure regime, and one extending well beyond the regime of measurements). The high-pressure behavior for both was constrained by requiring that the fourth derivative of the potential be small and that the high-pressure limit of *K'* be greater than 5/3 [45,46]. The high-pressure constraint on *K'* was enforced through non-linear optimization of an initial linear model. In this case, it is a choice to slightly misfit the pressure derivative of the bulk modulus for the highest-pressure data

as a necessary consequence of enforcing the smooth asymptotic behavior for $K'$. Here, physical insight is given more weight that the accuracy of a single data point at the extreme of the measurements.

The key findings of this section include the following: (1) Standard forms for equations of state based on global representations using series expansions of the Helmholtz energy can neither capture the details of NaCl behavior in regimes of measurement nor do they correctly extrapolate to high pressure. (2) Higher order global fits can be constructed that adequately fit all data, but even modest extrapolation is impossible. (3) In contrast, an LBF representation having three volume intervals (one extending well beyond measurements) can both better represent data and provides plausible (physics-based) behavior well beyond any possible measurement. (4) The choice of strain metric has no impact on the quality of the fit. (5) The behavior of representations in regions of extrapolation can be adjusted to match theory-based constraints. (6) The model size of the LBF representations are larger (a dozen or so parameters) compared to three for a global fourth-order fit. However, this is not a serious impediment to their use.

### 3.2. Equations of State for High Pressure Ice (Ice VII–X System)

#### 3.2.1. Data and Representations

Based on the success of local-basis-function representations in the previous section, attention is focused on the ice VII–ice X system to 450 GPa. The upper pressure limit of this representation is associated with the span of the density-functional (DFT) calculations. Equations of state at 300 K are constructed minimizing the root-mean-square (rms) misfit to 291 selected pressure–volume points from 16 independent datasets. While independently determined values for the bulk modulus have been reported and are discussed below, these data were not directly used as fitting constraints. Data and the numerical analysis are provided as scripts with the Supplementary Materials. Both experimental [15–17,21,33,47–57] and DFT studies [51,52] are considered. Results above 150 GPa are based solely on theory. The results in [15], that used an experimental method for pressure calibration, and measurements reported in [16], that deviate substantially from other work, are excluded in the current fits. Inclusion of these data increases the root-mean-square (rms) misfit but does not change the interpretations given below. No additional quality-of-data assessment is undertaken in the current analysis. Biases associated with systematic and random errors are likely reduced in the current meta-analysis of all data. The sample size may be sufficient to diminish the impact of small disagreements between merged datasets, arising from the use of different pressure gauges, volume measurement techniques, or experimental protocols that may affect biases associated with non-hydrostatic stresses. Pressure–volume data are plotted and compared with published equations of states [47,49,53,58] in Figure 2a.

To adequately explore the information contained in the pressure–volume measurements, an ensemble of representations was created based on differing assumptions for parameterizations and regularization. Examples are plotted in all panels of Figure 2, and parameters for these fits and associated misfits are listed in Table 3. The global fourth-order Eulerian representation, Equation (2), has the largest rms misfit. The LBF versions have reduced rms misfits. Although reduced misfit is expected for models containing more parameters, the global representation has greater systematic misfit and, as shown below, the LBF representations give results that better align with the lowest pressure bulk moduli data and the known complex phase behavior in the ice VII–X system. The LBF representation labeled "low structure" resulted from exploration (based on manual variation of knot placements) in which the goal was to find the smallest number of intervals with the least structure (smallest variations of $K'$). The impact of discontinuities in higher order derivatives of Helmholtz energy were tested by introducing knot multiplicity at boundaries that are manually aligned to match pressures for proposed higher-order phase transitions. One such fit labeled "transition informed" is reported and is further discussed below. "Agnostic" LBF fits were constructed using a sixth-order spline over 14 internal intervals to allow more than necessary flexibility in representing the equation

of state with no preconceptions of where higher-order phase transitions might occur. Two examples demonstrate differing levels of regularization. Additional LBF representations, created to explore hypothesized pressure dependences of the bulk modulus, are discussed and rejected in the following discussion. Regularization, based on minimizing the fourth derivative of Helmholtz energy, was applied to the LBF models. The damping parameter for regularization was varied to find an optimal trade-off between misfit and model smoothness. The tradeoff is illustrated in Figure 3 where the rms average of the Helmholtz energy fourth derivative is plotted against rms misfit of the pressure–volume data. An infinite number of solutions are possible as the damping parameter is continuously changed. As shown in the figure, larger rms values for the fourth derivative allow smaller misfit at the cost of possibly over fitting data. As greater smoothing is enforced (larger damping), misfit increases, eventually becoming unacceptably large. Three points marked on the diagram are associated with the three levels of misfit reported in Table 3. Acceptable solutions for damped solutions are typically chosen near the corner of the trade-off curve, representing a compromise between data misfit and the degree of smoothness.

**Table 3.** Equation of state parameters and root-mean-square (rms) pressure misfits ($P_{rms}$) for representations of high-pressure ice VII and X 300 K isotherm. Units for all parameters are shown. $V_o$ for ice VII is taken from Klotz (2017): 12.7218 cm$^3$/mol or 42.25 Å$^3$ at 300 K.

| Type of function | Parameters | | (rms) |
|---|---|---|---|
| | **Global-Basis-Function** | | |
| Fourth order/degree Eulerian Finite Strain | $K_o = 19.2$ GPa, $K_o' = 3.8$, $K_o'' = -0.09$ GPa$^{-1}$ | | $P_{rms} = 3.0$ GPa |
| | **Local-Basis-Function:** | | |
| | knots (dimensionless strain): | Coefficients (GPa cm$^3$/mole) | |
| "Agnostic" log Strain low damping Order: 6 (degree 5) | [−0.01, 0.02, 0.04, 0.06, 0.08, 0.10, 0.12, 0.15, 0.17, 0.20, 0.24, 0.27, 0.31, 0.36, 0.42] (first and last knots are repeated six times) | [−0.10, −0.07, 0.05, 0.40, 1.37, 3.75, 7.13, 12.5, 20.0, 30.7, 46.7, 72.5, 103, 157, 254, 379, 520, 647, 723] | $P_{rms} = 1.7$ GPa |
| "Agnostic" log Strain higher damping Order: 6 (degree 5) | [−0.01, 0.02, 0.04, 0.06, 0.08, 0.10, 0.12, 0.15, 0.17, 0.20, 0.24, 0.27, 0.31, 0.36, 0.42] (first and last knots are repeated six times) | [0.02, −0.03, −0.03, 0.23, 1.16, 3.55, 6.96, 12.3, 19.7, 30.4, 46.9, 71.5, 103, 157, 253, 375, 516, 645, 724] | $P_{rms} = 2.0$ GPa |
| "low structure" log Strain (seven intervals) Order: 7 (degree 6) | [−0.01, 0.08, 0.12, 0.16, 0.16, 0.24, 0.3, 0.42] (first and last knots are repeated seven times) | [−0.09, −0.04, 0.43, 2.21, 6.50, 17.5, 46.3, 93.1, 153, 258, 433, 595, 723] | $P_{rms} = 2.0$ GPa |
| "transition informed" log Strain (seven intervals) Order: 7 (degree 6) | [−0.01, 0.12, 0.16, 0.16, 0.21, 0.23, 0.26, 0.42] (first and last knots are repeated seven times) | [0.10, −0.27, 0.26, 2.86, 10.6, 25.6, 57.3, 104, 153, 263, 396, 556, 722] | $P_{rms} = 1.7$ GPa |

Fractional deviations from the "low-structure" representation of the pressure volume data and selected literature equations of state are plotted as a function of the logarithm of pressure in Figure 2b. As shown in this panel, the fractional deviations are relatively uniform as a function of pressure with a standard deviation near 5%. The highest-pressure points based on theory have greater precision with unknown accuracy but appear to merge into the more scattered measurements with little apparent offset or change in slope.

In Figure 2c (full pressure range) and d (lower pressure regime), predictions of the isothermal bulk modulus based the current equation of state are compared to reported adiabatic bulk modulus

based on Brillouin acoustic measurements [31,59–61]. The difference between adiabatic and isothermal moduli is small in the high-pressure regime and is currently ignored. Although the conversion from sound speeds to elastic moduli requires knowledge of the density, the differences in density between the current representation and those used in the original studies had negligible impact at the scale of these figures. The measurements to 8 GPa reported by Shimizu et al. [60] for single crystals grown in equilibrium with liquid water provide plausible determinations of the bulk modulus. Other studies relied on longitudinal and transverse wave measurements in polycrystalline material with unknown fabric in a non-hydrostatic environment. In spite of the cubic symmetry for the high-pressure ices, such data may suffer from systematic biases.

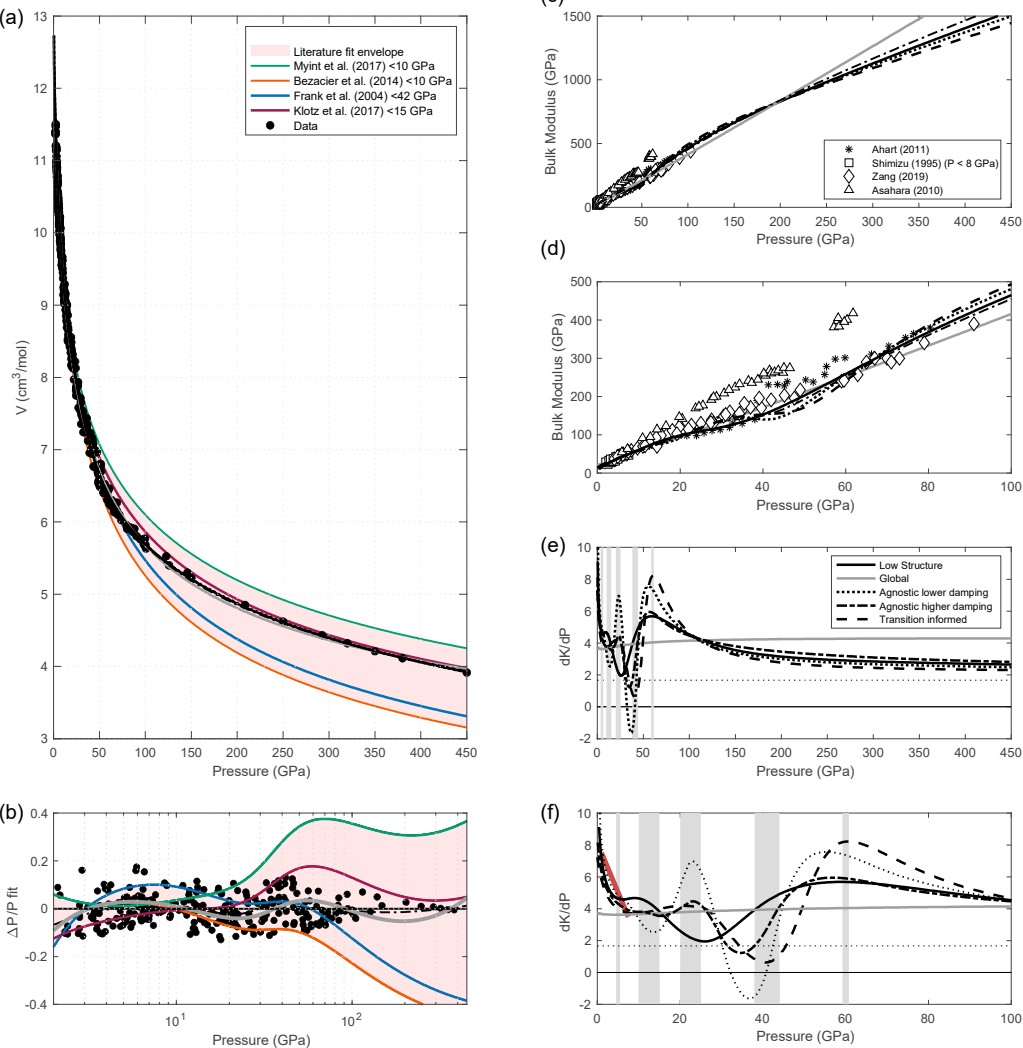

**Figure 2.** Data and equations of states of ice VII and X at 300K. In all panels, predictions based on the "low structure" local-basis-function (LBF) are shown with thick solid lines, the dotted and dot-dashed lines are the "agnostic" LBFs with lighter or heavier damping, the dashed lines are "transition informed", the gray lines are the global fourth-order fit. (**a**) Specific volume as a function of pressure. All measurements are represented with solid circles. Prior fits are identified in the legend. (**b**) Residuals relative to the preferred LBF representation as a function of the logarithm of pressure. (**c**,**d**) Bulk moduli as a function of pressure. All symbols are identified in the legend. (**e**,**f**) Pressure derivative of the bulk moduli for the four current representations. The 5/3 limit for high-pressure behavior of K′ in represented as a horizontal dotted line. The behavior of K′ based on a quadratic fit to Shimizu et al. [60] is shown for pressures between 2 and 8 GPa a thick line. Vertical bars indicate identified pressure ranges (Table 1) for higher order transitions.

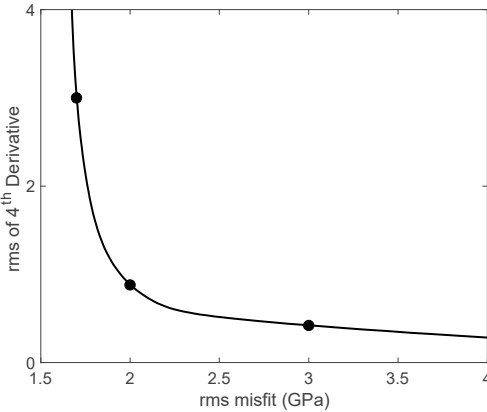

**Figure 3.** Tradeoff between root-mean-square (rms) data misfit and model smoothness as measured by the root-mean-square average of the fourth derivative of Helmholtz energy. The line defines the locus of solutions obtained by varying the damping factor for regularization. Circles mark points with rms misfits of 1.7 GPa, 2 GPa, and 3 GPa that are associated with fits listed in Table 3.

Pressure derivatives of the bulk modulus based on our representations are shown in Figure 2e (full pressure range) and f (lower pressure regime). Also shown between 2 GPa and 8 GPa is the behavior of $K'$ as constrained by a quadratic fit to the single crystal measurements of Shimizu et al. Vertical bands in these panels indicate the pressure ranges for the suggested higher-order transitions listed in Table 1.

### 3.2.2. Discussion

The following discussion progresses through considerations of (1) the pressure–volume data, (2) the elastic bulk moduli determinations, and finally, (3) how variations of the pressure derivative of the bulk modulus correlate to pressure ranges where higher-order transitions likely occur. In the absence of transitions and as previously documented in Figure 1 for NaCl, values for the pressure derivative of the bulk modulus, $K'$, can range from more than 6 at low pressure to less than 3 at the highest pressures. The second derivative, $K''$, is usually small and negative. Possible changes of ice's compressive behavior related to higher order transition as summarized in Table 1 include the following ideas. A possible distortion in the bcc lattice around 15 GPa is suspected to be the expression of the locking of OH-rotational disorder which could change the rate of stiffening under compression [51]. Changing derivatives of the bulk modulus near 25 GPa might reflect the transition from ice VII to ice VII′ associated with the start of dynamic translational proton disorder by tunneling and thermal hoping. The onset of the hydrogen bond symmetrization, with the transition towards H-translational disorder [51], should have an effect on compression beyond 40 GPa. Furthermore, a broad minimum in $K'$ would be the expected behavior for a second order transition between ice VII and a slightly higher density ice X phase.

In prior analyses of individual datasets, equations of state were constrained by data spanning restricted ranges of pressures. The envelopes of such regional fits, extrapolated to cover the entire range of pressure, are illustrated in Figure 2a,b. The various fits are essentially indistinguishable below 50 GPa (Figure 2a). Even at the scale of the residuals (Figure 2b), data scatter is as large as the envelop bounds. Unsurprisingly, predictions in regimes of extrapolation diverge and do not accurately match the highest-pressure DFT results.

The adequacy of representations as measured by the rms misfit to data is not immediately obvious in examination of just the pressure–volume relationships shown in Figure 2a,b. However, the trade-off curve shown in Figure 3 indicates that an rms misfit near 2 GPa is optimal. On this basis, the global fourth-order representation clearly underfits the data, while the low damping agnostic fit and the transition informed fit may overfit the data (as further discussed below).

Greater discrimination between equation of state representations is expected at the level of the derivative properties: the bulk modulus and its pressure derivative (all panels on the right side of

Figure 2). For pressures below 8 GPa, *K* and *K′* values determined from sound speed measurements are replicated in the LBF fits, indicating that the underlying pressure–volume data are sufficiently accurate and that the analysis can extract higher order derivatives of Helmholtz energy. In contrast, all global fits, both from prior literature and the current fourth-order, do not match the low-pressure behavior for *K′*. In order to approximate the volume trends over the entire range of pressures, all global fits require a nearly constant and low (~4) value for *K′* since such global representations have insufficient flexibility to accommodate both behavior at low pressure and at high pressure. This mirrors the findings in the previous section with the effort to represent the NaCl equation of state. At the highest pressures, where only theory provides constraints, the LBF representations smoothly approach the Thomas–Fermi limit. No global representation provides correct limiting behavior. Furthermore, the current fourth-order global representation shows unphysical behavior with *K′* increasing from 3.8 near zero pressure to a nearly constant value of approximately 4 at the highest pressure.

In the critical range of higher-order transitions for ice between 15 GPa and 60 GPa, the measured bulk moduli show disagreement between the independent studies and the LBF representations Whether these differences are caused by systematic errors in measurement, differences in non-hydrostatic stresses, or are indicative of material properties that vary with the time-scale of measurement, remains unresolved. A hypothetical relaxation time for a phase transition or for proton mobility that is large compared to the sub-nanosecond periods of acoustic wave measured in Brillouin experiments could rationalize the observation that the static moduli are smaller than the moduli based on sound speeds. Equation of state fits required to match the Brillouin-based bulk moduli in this pressure regime were found to systematically misfit the pressure–volume data. In contrast, the bulk moduli beyond 60 GPa [59,61] converge to a common pressure dependence that is consistent with predictions based on the LBF fits.

Focusing on *K′* in Figure 2e,f, trends for its pressure dependence separate into three zones. In the low-pressure regime *K′* decreases from a value near 6.5 at 2.2 GPa (similar to values for ice V and ice VI [40]) to about 4 for pressures between 10 to 15 GPa. As shown in the figure, this normal trend is in agreement with the Shimizu et al. determinations to 8 GPa. Above 60–70 GPa, *K′* also shows normal behavior, an asymptotic decrease with pressure. In contrast, the intermediate pressure range, the previously identified regime of higher-order transitions, shows the most anomalous behavior for *K′*. All LBF representations show a minimum between 30 GPa and 40 GPa. The "agnostic" fits have shoulder maxima on the sides of the minimum values. One "agnostic" fit has an optimal misfit of 2 GPa (based on Figure 3). The other "agnostic" fit, with smaller rms misfit (that appears on the overfitting side of the trade-off curve), predicts that *K′* dips below zero resulting in a local minimum of *K* as shown in Figure 2d. The "low structure" representation has a shallower minimum and no shoulder maxima around 25 GPa for the same (optimal) 2 GPa misfit. Since *K′* is negative in the transition region only for the least damped "agnostic" solution, the bulk modulus does not generally "soften" (decrease at higher pressure). Instead, it simply fails to increase normally with pressure. The "transition informed" fit, created to better align features of *K′* with identified pressures of transitions, has interval boundaries near the transition from dominant rotational disorder to more translational disorder around 15 GPa and the start of hydrogen symmetrization with the increase of translational disorder above 40 GPa, with a full transition to ice X above 60–80. As shown in Table 3, the misfit for the "transition informed" representation is reduced at the cost of a higher rms average for the fourth derivative of Helmholtz energy. Whether this represents over fitting data remains to be determined.

The key findings of this section reinforce those associated with the effort to represent NaCl. Conventional equation of state global representations can provide approximations to all pressure–volume data at the expense of predicted bulk moduli that are not in accord with low pressure measurements and exhibit incorrect high-pressure limiting behavior. Local-basis-function representation for high-pressure ice are able to better satisfy these strong side constraints. However, for the ice VII–X continuum, in the regime of higher-order phase transitions, an infinite number of representations are possible that fit the data within plausible bounds. The ensemble reported here

provide examples of trade-off possibilities between misfit to data and allowable structure of the fits. Representations can have features of *K'* aligned with expected transitions on the basis of a priori constraints although such close alignment is not required by the pressure–volume data alone. The most robust features of all members of successful fits are a decrease in *K'* from normal values near 15 GPa to a minimum in the 25 to 40 GPa range followed by an increase to 60–70 GPa followed by normal asymptotic decreases at much higher pressures. No fit required to follow measured bulk moduli in the 25 to 60 GPa range provided an adequate representation of the pressure–volume data. Whether this is a result of errors in data or is an indication of a rate-dependent process remains undetermined.

## 4. Discussion

A representation of Helmholtz energy at 300 K for ice VII–X was developed in order to better understand the complexity of high-pressure ices that exist as mineral phases in this solar system and beyond. Questions of origins and evolution require knowledge of an underlying equation of state. The results of this investigation are summarized in the concluding section. In order to accomplish the principal goal, coverage of a broad set of concepts and analysis was necessary. Topics include a foundational review about the nature of numerical representations for equations of state, an exposition describing a new framework for representations, and articulation of important details associated with the new framework. Both interpolation and inverse problems are widely recognized challenges in data assimilation. Methods, previously found useful in other fields, are adapted and modified to meet the unique requirements of Helmholtz energy representations.

The underlying premise is that equations of state must provide coherent empirical representations of all measurement-constrained and theory-based knowledge of a material. The previous generation of parameterizations under performed in this task as a result of a bias towards using physics-blind models with insufficient flexibility. This lack of flexibility resulted in discussions that shifted to numerical issues such as "What constitutes the best strain metric?" and "Which order expansion is preferred?". The current analysis refocuses attention. The behavior of an equation of state should be physics-based and basis-function independent. The total number of model parameters required is less important than how well the representation matches the constraining information. Finding the best representation involves determining the appropriate relative weighting of data from varied sources, assessing the quality of results based on theory, and applying suitable side-constraints based on the best physical insight. Decisions based on these ideas were specifically expressed during the construction of equations of state for NaCl and high-pressure ice.

The new equation of state analysis is a seminal step beyond the earlier generation of Helmholtz energy-based parameterizations that use polynomial series expansions of an arbitrarily defined strain metric. Here, a single global polynomial is replaced with flexible local piece-wise polynomials in the form of b-splines. The new local-basis-function representations are more than simple spline fits of data since their creation is embedded in an inverse theory framework of parameter estimation by Tikhonov inversion that requires incorporation of physics-based constraints and insight. An important characteristic of inverse theory solutions is the ability to explore features shared by all possible representations of data. A key strength of this approach, demonstrated in the effort to represent ice, is the power to ask both what model features are required by data and conversely what are data requirements to adequately delineate model features.

## 5. Conclusions

A large body of measurements and theory for ice VII–X were assimilated into an equation of state covering pressures up to 450 GPa. Selected pressure–volume data from 14 experimental studies and two theoretical reports were then accurately represented. In addition, separately measured derivative properties (the bulk modulus and its pressure derivative) were replicated both in low-pressure and high-pressure regimes, indicating that the underlying pressure–volume data are sufficiently accurate and that the current analysis can extract higher order derivatives. Anomalies in the bulk modulus (and

its derivative) were found to roughly align with proposed higher-order transitions at intermediate pressures. However, in this range, bulk moduli determined using GHz-frequency Brillouin experiments did not match the static determination. The differences between results from different groups leaves open the possibility that systematic experimental errors remain large. Alternatively, a relaxation time-constant for the underlying high-order phase transition or for proton mobility may be large compared to the frequencies used in acoustic measurements. The pressure–volume data can resolve variations of the second derivative of the bulk modulus with pressure. However, discontinuities in $K''$ and/or the precise locations of features in the pressure derivative of the bulk modulus are not currently resolvable. Improved high-pressure determinations of derivative properties could provide better constraints for the next-generation representations. More generally, the tools and methodologies developed here can be broadly applied in other high-pressure equation of state studies.

Articulation of the necessary background for b-splines and for parameter estimation is placed in Appendix A. A small toolbox containing five functions and example scripts written in the MATLAB numerical environment are also provided with the Supplementary Materials. MATLAB is convenient for exploration of the ideas presented in this paper. The open source environment OCTAVE is compatible with the provided MATLAB functions and scripts and can be downloaded for no cost. PYTHON and FORTRAN implementations can easily be created.

In the provided MATLAB scripts, the analysis requires four steps: (1) load data, (2) set options, (3) fit data, (4) display results. This provides a straightforward workflow that encourages exploration of how modifications of assumptions provide differences in the resulting equation of state. It is important to try modifications of all adjustable elements in order to explore the parameter sensitivity and quality variations of fits.

**Supplementary Materials:** The following are available online at http://www.mdpi.com/2075-163X/10/2/92/s1, computer software files.

**Author Contributions:** Conceptualization, J.M.B. and B.J.; methodology, J.M.B.; software, J.M.B.; validation, J.M.B. and B.J.; formal analysis, J.M.B. and B.J.; writing—original draft preparation, J.M.B. and B.J.; writing—review and editing, J.M.B. and B.J. All authors have read and agreed to the published version of the manuscript.

**Funding:** This work is partially supported by the NASA Solar System Workings Grant 80NSSC17K0775 and the NASA Astrobiology Institute through the Icy Worlds and the Titan and Beyond; Habitability of Hydrocarbon Worlds (08-NAI5-0021 and 17-NAI8-2-017). B.R.J.'s research was partially supported by an appointment to the NASA Postdoctoral Program at the University of Washington, administered by Universities Space Research Association under contract with NASA.

**Acknowledgments:** Long running discussions with E. Abramson improved the current presentation. We thank Philip Myint for his encouragement to undertake this work. The authors also thank J-A. Hernandez for fruitful discussions on DFT calculations of ice. J.M.B. expresses gratitude for the influence Orson L. Anderson had on his career. Anderson's research, helping to define the field of mineral physics, provided an impetus for J.M.B., as an undergraduate student, to switch from atomic physics to geophysics. Later, J.M.B. had the privilege to work with O.L.A. in the founding of the Mineral Physics Group within the American Geophysical Union. O.L.A.'s record of scholarship, his enthusiasm, and his support for others are lasting legacies of his career.

**Conflicts of Interest:** The authors declare no conflict of interest.

## Appendix A. B-Spline and Inverse Method Details Related to Equation of State Representations

### *Appendix A.1. B-Spline Basis Functions*

Fundamentals of b-splines are covered in detail in de Boor [29]. Here, a few points are highlighted in association with the use of b-splines in equation of state representations. B-splines are piece-wise polynomials of specified order, on intervals defined by the interval bounding knots. In polynomial representations, $x^3$ is both a third degree and third order polynomial. However, using accepted convention, a *kth*-order spline is associated with polynomial of degree *k-1*. To avoid confusion in the following discussion, the spline order is parenthetically followed by explicit articulation of the underlying degree of an associated polynomial.

Knots are points along the independent variable ($x$) axis that define interval boundaries. For a knot sequence ($t_1, t_2, \ldots, t_{p+k}$), where $p$ is the number of coefficients (the model parameters) and $k$ is the order of the b-spline (degree $k$-1), a recursion relationship (Equation (A1)) defines the spline basis function, $B_{j,k}(x)$, where $x$ lies between the first and last knot and the first order basis function $B_{j,1}$ is equal to 1 for $x$ within the interval between $t_j$ and $t_{j+k}$ and 0 otherwise.

$$B_{j,k}(x) = \frac{x - t_j}{t_{j+k-1} - t_j} B_{j,k-1}(x) + \frac{t_{j+k} - x}{t_{j+k} - t_{j+1}} B_{j+1,k-1}(x) \tag{A1}$$

Analytic values for basis functions, $B_{jk}$ evaluated at specified locations, are numerically obtained and represented as arrays with a column count equal to the number of spline coefficients (model parameters) and a row count equal to the number of locations of the function evaluation. The underlying algorithms are robust and available as functions/subroutines in all numerical environments. The use of b-spline basis functions requires no more concern on the part of the user than the use of any standard numerical functions (i.e., trigonometric or logarithmic functions). However, in order to develop new equations of state, the user needs to know how to create an acceptable knot series for a specific application. The knots ($t_i$) used to construct the spline basis functions must meet several requirements. (1) The total number of knots must equal $k + p$. (2) Each repetition of a knot reduces by one the number of continuous derivatives at the interval boundary. (3) The first and last knot are repeated $k$-fold times in order for arbitrary data to be fit. Further discussion of how knots are chosen is given below.

*Appendix A.2. Evaluation of B-Spline Representations*

The evaluation of $y = f(x)$, where function $f$ is a b-spline representation, is numerically represented:

$$y_l = \sum_{i=1}^{p} B_{j,k}(x_l) m(i) \tag{A2}$$

where data are paired values ($x_l$ and $y_l$), vector $\overline{m}$ contains $p$ spline coefficients (the model parameters), and $B_{j,k}(x_l)$ values are determined using Equation (A1). Using vector and matrix notation Equation (A2) is simply a linear equation of the form:

$$\overline{y} = \overline{\overline{B}}\,\overline{m} \tag{A3}$$

which leads naturally and advantageously to spline coefficients, $\overline{m}$, being determined through the inverse solution of a linear system (further discussed below). As written in Equation (A2), the determination of a set of modeled $\overline{y}$ values requires that a (possibly large) matrix of basis functions be determined and held in computer memory. The Cox–de Boor algorithm is implemented in all numerical environments and allows Equation (A2) to be more efficiently evaluated (not holding the matrix $\overline{\overline{B}}$ in memory) which reduces computer CPU and memory requirements.

Evaluation of derivative or integral properties of a b-spline representation simply requires analytic determination of the appropriate derivative or integral values of the polynomial basis functions $B_{j,k}(x_l)$:

$$DB_{j,k}(x) = \frac{k-1}{t_{j+k-1} - t_j} B_{j,k-1}(x) - \frac{k-1}{t_{j+k} - t_{j+1}} B_{j+1,k-1}(x) \tag{A4}$$

followed by a solution using the form shown in Equation (A2).

*Appendix A.3. Details of B-Spline Knots and Control Points*

As an example of knots for a ($k = 4$) fourth-order (third degree) b-spline on a domain spanning from 1 to 10, an arbitrary chosen knots sequence containing the nine elements [1, 1, 1, 1, 5, 10, 10, 10,

10] divides the domain into two intervals (1 through 5 and 5 through 10). Bounding knots are repeated *k* times. Based on the relationship that the number of knots is equal to the sum of the number of spline coefficients (model parameters) and the spline order, five spline coefficients are required to describe the two associated cubic polynomials. While a single third-degree polynomial has four possible coefficients, the additional polynomial associated with a second interval requires one additional coefficient since the matching boundary conditions fixes the other three parameters. The sequence [1, 1, 1, 1, 4, 7, 10, 10, 10, 10] has three intervals using 10 knots that require six spline coefficients. The spacing of intervals is an arbitrary choice to meet needs in a particular application. An alternate sequence with 10 knots having different intervals is [1, 1, 1, 1, 2, 4, 10, 10, 10, 10]. The sequence [1, 1, 1, 1, 2, 4, 4, 10, 10, 10, 10] (with one interior knot repeated) contains 11 knots and requires seven coefficients. The underlying representation is then allowed to have a discontinuous second derivative at location 4.

An important distinction is made between splines as a tool to interpolate precise data and splines as a method to represent, in a least square sense, data with uncertainty. In the historical development of spline methods, data (pairs of dependent and independent variables) were assumed to have high precision and basis functions were constructed to provide accurate interpolation between the data. In this context, the number of knots is chosen such that the number of required model parameters exactly matches the number of constraints (the available data). Knot locations are then chosen such that at least one data point is associated with each interval. For such an exactly constrained problem, the locations of the independent variables are labeled "control points". The resulting model parameters are then directly associated with the specified control points. However, the model, given as a list of the knots (different from the list of control points) and a list of model parameters, does not require that control point locations be specified. In the least square fitting of more data than model parameters, with representations that are constructed through choice of knot locations (not a choice of data locations), the control point locations are ill defined other than that model parameter locations sequentially progress across the span of the knots with a few more model parameters required than intervals. This discussion is provided since the standard presentations of splines tend to emphasize a perspective centered on control points.

*Appendix A.4. Local-Basis-Function Equations of State Representations*

A simplified equation of state application using b-splines is illustrated in Figure A1. Panel (a) on the left shows Helmholtz energy as a function of volume. The corresponding pressures (the negative derivative of Helmholtz energy) are shown on the right side in panel (b). An energy minimum on the left is associated with zero pressure on the right. Pressures increase as volumes decrease from equilibrium. Pressure becomes negative in the expanded state but tends back to zero for sufficiently large volumes.

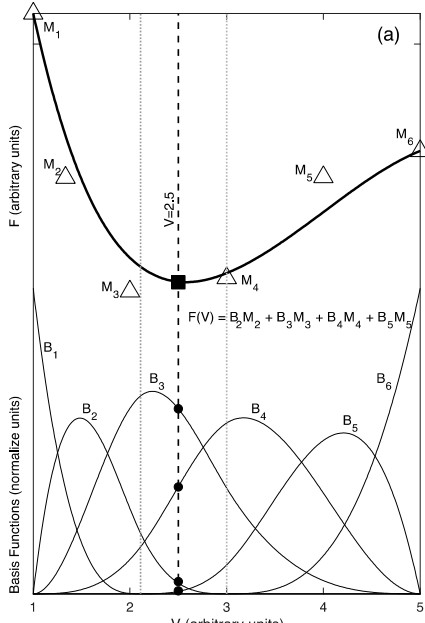
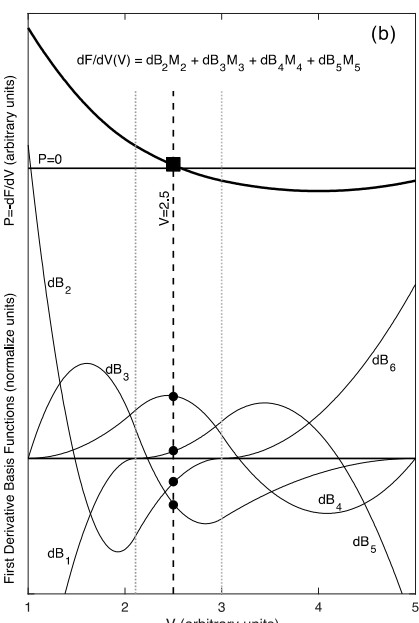

**Figure A1.** Equation of state example using a cubic b-spline representation on three intervals. Panel (**a**) (left side): Helmholtz energy (F) as a function of volume (V) is shown as a thick line. Panel (**b**) (right side): Pressure, as the negative of the first derivative of Helmholtz energy, is shown. Vertical dotted lines show the internal spline boundaries between three intervals of volume. B-spline basis functions are shown at the bottom of each panel with labels $B_i$ and $dB_i$. The six model parameters (labeled $M_i$) required to represent the function are shown in panel (a), plotted at their control point locations. Note that the control point locations are implicit in the distribution of the intervals and are not separately tabulated in the model description. The evaluation of the function at a specified volume (vertical dashed line) is illustrated in each panel where the function is given as a sum of the product of basis function values and model parameters. The evaluation of the functions for any other value of volume is undertaken by the same linear analysis. The model parameters, in units of energy, can be determined through a fit of pressures and volumes based on the linear relationships as shown on the right side.

In this example, three intervals are chosen for a b-spline representation using cubic basis functions (a fourth-order spline for energy). The ranges of the internal intervals are marked by the vertical dotted lines. If three independent cubic polynomials were used for these intervals, 12 separate basis functions and parameters would be required (four degrees of freedom for each cubic polynomial). However, matching the function and its first two derivatives at the two internal boundaries reduces the problem to six degrees of freedom—two additional basis functions are required beyond the four that otherwise would be associated with a single global cubic polynomial. Given specified intervals and spline order, the b-spline basis functions are exactly defined and numerically determined by a subroutine call. The behavior of the six b-spline basis functions is shown at the bottom of each panel.

The b-spline basis functions are non-zero over subdomains of the full model space, extending over *k* intervals across some interval boundaries. Basis function in the right panel are clearly derivatives of basis functions plotted in the left panel. Model parameters, determined by fitting data on the right side, are plotted in the left panel. The volumes associated with model parameters (the control points as previously described) are not directly aligned with the interval boundaries; instead they are implicit in the distribution of intervals and do not figure into evaluations of the splines. In the limit of small intervals, the model parameters are equal to the function being represented within each interval.

Interpolation using b-splines is a weighted average of neighboring model parameters, the basis functions provide the normalized weights. As shown in Figure A1, both the model and the derivative of the model are evaluated as a linear problem: a sum of basis function values times model parameter values. The same model parameters are used for calculation of the function and for calculation

of its derivatives. This provides a path to determine the Helmholtz energy by collocation, using measurements of its derivatives. Implicit is the need to set an integrating constant. The inverse problem, determining model parameters from data, is further discussed below.

*Appendix A.5. Defining Equation of State Properties with Derivatives of Helmholtz Energy*

In the current framework, equation of state properties are numerically evaluated using analytic representations for the underlying quantities. The necessary relationships associated with derivatives of Helmholtz energy that define pressure, the bulk modulus, and its pressure derivative are summarized. Since Helmholtz energy is parameterized as a function of strain $F(\eta, T = 300\ K)$, the relationship of $\eta$ as a function of volume figures into the thermodynamic quantities. Derivatives of strain with volume are analytically determined from the variously defined functional forms and derivatives of energy are analytically calculated from the local-basis-function representation. These derivatives are denoted using subscripts for the derivative level:

$$\eta_1 = \frac{d\eta}{dV},\ \eta_2 = \frac{d^2\eta}{dV^2}\cdots$$

and

$$F_1 = \frac{dF}{d\eta},\ F_2 = \frac{d^2F}{d\eta^2}\cdots$$

Through application of the chain rule for differentiation, pressure is then calculated as a product of derivatives:

$$P = -\frac{dF}{dV} = -\frac{dF}{d\eta}\frac{d\eta}{dV} = -F_1\eta_1 \tag{A5}$$

In order to calculate the bulk modulus and its pressure derivative, derivatives of pressure with volume are required

$$P_1 = \frac{dP}{dV} = -\eta_1{}^2 F_2 - \eta_2 F_1$$

And

$$P_2 = \frac{d^2P}{dV^2} = -\eta_1{}^3 F_3 - 3\ \eta_1\ \eta_2 F_2 - \eta_3 F_1$$

Using these derivatives, the bulk modulus is

$$K = -V P_1 \tag{A6}$$

and its pressure derivative is

$$K' = \frac{dK}{dP} = -V\left(1 + \frac{P_2}{P_1}\right) \tag{A7}$$

From these relationships it is apparent that pressure and the bulk modulus are determined as linear combinations of strain basis functions while the pressure derivative $K'$ requires a non-linear combination of strain basis functions.

*Appendix A.6. Inverse Techniques to Find Model Parameters*

As described in Aster et al. [33], parameter estimation based on inverse theory is a foundational problem in diverse fields where data assimilation and interpretation are required. Here a selection of ideas is presented that directly bear on equation of state studies. As shown above, with Helmholtz energy expanded in a strain metric, both pressures and bulk moduli as a function of volume can be expressed using linear relationships. A general linear problem, associated with any equation of state application, is given as:

$$\bar{\bar{B}}\bar{m} = \bar{d} \tag{A8}$$

where the array $\bar{\bar{B}}$ on the left-side contains values for the basis functions evaluated at the independent variable locations associated with the dependent variables $\bar{d}$ given in the right-most column vector, and $\bar{m}$ contains the model parameters. For $m$ model parameters and $n$ data, $\bar{\bar{B}}$ is an $n$ by $m$ matrix of values. In the case of conventional equations of state fitting, the basis functions are the polynomial values of strain shown on the right side of Equation (2) (main text) or the fractional powers of (V/V$_o$) shown on the right side of Equation (4) (main text). For local-basis-function representations, the b-spline basis functions evaluated for each independent variable location, are placed in $\bar{\bar{B}}$.

Following standard numerical analysis, the least square solution for a linear problem (Equation (A8)) for parameters $\bar{m}$ is

$$\bar{m} = (\bar{\bar{B}}^t \bar{\bar{B}})^{-1} \bar{\bar{B}}^t \bar{d} \tag{A9}$$

where superscript $^t$ denotes the transpose operation and the negative one power implies determination of a matrix inverse. Equation (A9) can usually be solved if more data than model parameters are available (not rank-deficit). However, even with an adequate amount of data, solutions may be poorly-conditioned when data do not adequately span the parameter space of the model. The typical example of a poorly-conditioned solution is the exercise of trying to fit a linear distribution of data using a higher-order polynomial. Problems can also arise in constraining model parameters for a local-basis-function representation if data are not adequately distributed in regimes containing measurements or in regions of extrapolation.

To overcome problems of poorly-conditioned or rank-deficit inverse problems, additional side constraints are needed and are included through regularization. A common form of regularization is to make the model "smooth" by minimizing its second derivative. More generally, based on physical insight in equation of state representations, a higher-order derivative of Helmholtz energy is minimized, as described in the main text. Basis functions for the specified derivative of energy are added to the array $\bar{\bar{B}}$ and associated zeros are added to the data vector $\bar{d}$:

$$\begin{bmatrix} \bar{\bar{B}}_d \\ \lambda \bar{\bar{B}}_r \end{bmatrix} [\bar{m}] = \begin{bmatrix} \bar{d} \\ 0 \end{bmatrix} \tag{A10}$$

where $\bar{\bar{B}}_d$ are the basis functions associated with data, $\bar{\bar{B}}_r$ are the basis functions for regularization (a chosen derivative of Helmholtz energy), and $\lambda$ is an adjustable (damping) factor to weight the influence of the regularization (small value for no influence and large value to strongly enforce the regularization). The data vector on the right side of Equation (A10) contains both measurements, $\bar{d}$, and as many zeros as rows of regularization in array $\bar{\bar{B}}_r$. The idea is that $\bar{\bar{B}}_r\bar{m}$ should be as close to zero as possible, while simultaneously requiring the model to adequately fit the data. In areas of sparse or no data, regularization controls the nature of the fit. The advantages of a least square solution of Equation (A10) includes an ability to suppress unnecessary structure (overfitting) and to provide constraints in regimes without data. Linear optimization of pressure–volume and bulk modulus–volume data with regularization based on a user-specified derivative of Helmholtz energy is implemented in the numerical toolbox included with the Supplementary Materials.

In order to directly fit measurements that cannot be expressed as a linear combination of model parameters (i.e., derivatives of the bulk modulus with respect to pressure), non-linear optimization of model parameters is required. The form of the non-linear Tikhonov problem is given as:

$$\begin{bmatrix} \bar{\bar{C}} \\ \lambda \bar{\bar{Q}} \end{bmatrix} [\delta \bar{m}] = \begin{bmatrix} \delta \bar{d} \\ -\lambda \bar{\bar{Q}} \bar{m}_o \end{bmatrix} \tag{A11}$$

where $\overline{m}_o$ is an initial guess for the model, $\delta\overline{m}$ are increments to $\overline{m}_o$ that reduce data misfit, and $\delta\overline{d}$ are deviations between data and predictions based on $\overline{m}_o$. Data can be a combination of equation of state measurements including pressures, bulk moduli, or derivatives of the bulk modulus. The array $\overline{\overline{C}}$ contains derivatives of the model predictions with respect to model parameters and the array $\overline{\overline{Q}}$ contains derivatives with respect to model parameters of any regularization quantity to minimize. The least square solution of Equation (A11) gives the increments $\delta\overline{m}$ to $\overline{m}_o$ that provide a better representation of data subject to the side constraint. Using an improved model $\overline{m}_o$ after each step gives results that usually converge after a few iterations. Non-linear optimization that includes constraints on the pressure derivative of the bulk modulus is implemented in the numerical toolbox included with the Supplementary Materials.

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
