# Peer review of "Local-Basis-Function Equation of State for Ice VII–X to 450 GPa at 300 K"

_minerals, doi:10.3390/min10020092_

Round 1

Reviewer 1 Report

The manuscript by Brown and Journaux describes theoretical study on equation of state for ice VII-X in a pressure regime extending to 450 GPa at 300 K using local basis functions in the form of b-splines. Using pressure-volume data from 14 experimental and 2 theoretical studies, authors measured derivative properties and replicated both in low-pressure and high-pressure regimes, indicating that the underlying pressure-volume data are sufficiently accurate. In addition, authors provided an example MATLAB script which is useful for exploration of how modifications of assumptions provide differences in the resulting equation of state. The manuscript was easy to read, interesting and deserves publication in the journal with only a few minor revisions.

The number of references in the text and the list of references do not match (in the text - 58, in the list of references - 54). It would be nice to make Figure 1 and Figure A1 as Figure 2 with the labels (a) (b) (c). In Figure 1 density units g/cc?

Author Response

We thank the reviewer for the attention to the details and the review.

The reference list has been revised and repaired - at some point the last page of the submission with the missing references was lost. Although it would be possible to reorganize Figures 1 and A1 as the reviewer recommends, we believe this would disrupt the organization of the paper and our effort to separate science (Figure 1) from the important technical details (Figure A1). We have chosen to stay with our current organization We have changed the label to a fully metric unit (Mg/m3) - same numeric value.

Reviewer 2 Report

The concept presented in this manuscript, that b-spline type local-basis-function can be used for expression of the equation of states, seems innovative, robust, and widely applicable as a substitution of conventionally used EOSs. Furthermore, this concept may be straightforwardly extended to a higher-dimensional dataset, such as P-T-V equation of states using some b-spline surface function. I suppose this manuscript could be a milestone, if it really substitutes the conventional EOS. I wish I had more time to check the robustness and more carefully follow the procedure, but unfortunately, only one week is allowed for this journal. Honestly speaking, this manuscript should be published through more well-reviewed process (means, in a different journal). In spite of that, at a first glance, I find the importance of this manuscript and recommendable for publication mostly as it is. Only a few minor issues could be indicated at the moment as follows.

Recently found ice XVIII (Millot et al., Nature, 2019, 10.1038/s41586-019-1114-6) should be included in a review of ice VII-X studies. It could be considered as post-ice X phase, having fcc oxygen sublattice. I could not properly follow (probably just due to the limited time to review) what authors actually did for the analysis ‘transition informed’. Please describe more details if it is not in the current manuscript. Table 1 seems very useful for readers. Could you provide the name of the first authors in ref list? (like Journaux [6]) This way could be a more memorable style. Figure 2 legend. Ashara (2016) may be Asahara (2016). Figure 2 and Table 3. It could be better to show the correspondence of types of LBF and style of lines, maybe in an additional legend in (e) and (f).

For my personal interest, it would be very helpful if authors provide a tutorial of MATLAB code in a web-site.

Author Response

We thank the reviewer for the thoughtful comments

The Millot et al paper is now cited and our summary of their findings can be found at line 90 of the revised manuscript The "transition informed" representation is described at line 508 in the revised manuscript We have added the names to the list in Table 1. Asahara (2016) is corrected in the Figure An additional legend is added in Figure 2 per reviewer suggestion The first author's personal web site has a continuing update of software and examples.

Reviewer 3 Report

This is an excellent paper, and I am very impressed by the quality of the writing, the clarity of its organization, and the care the authors have clearly taken to ensure that the mathematical development is (as far as I can tell) error-free.

I should note that I have not reproduced any of the calculations or plots, nor have I tested the Matlab routines. That said, I think the author-provided plots and discussion, in and of themselves, demonstrate that the methods, calculations, and interpretations are sound.

The authors have also done an excellent job of reviewing the relevant, and available, experimental results; I have not been able to identify any studies that are not included. Furthermore, the authors are judicious in their data selection and I do not anticipate objections from the many workers who have conducted experimental studies on high-pressure ices. (That's not a small feat because there has been a lot of prior work of varying quality, coupled with strong covariance between K and K', and this further demonstrates the value of the approach presented here.)

My only concern with the present work is that even with the provided Matlab files (which the authors test and note can also be run on the freely-available Octave software), the current approach will prove to be too cumbersome for widespread adoption. Then again, the author's aren't seeking to make a wholesale replacement of commonly used methods, but are, instead, specifically trying to address challenges on phases like high-pressure ices for which the need for reliable extrapolation is high, and it may be that in such cases other workers will indeed adopt these techniques. 

And, the authors provide exceptionally clear and sound discussion of the Helmholtz, their consideration of strain representations, comparison with previous EoS developments and, most notably, their use of b splines. Indeed, I believe other workers could follow (and implement) the fitting techniques with no prior experience in using these methods.

Regardless, I don't view that as an obstacle for publication as my pessimism about the actual impact of this work may well be wrong. And, I think the authors should be commended for producing what I view as a publication-ready submission. That is, I truly believe that this paper is ready to be published as-is, and that only time will tell how much of an impact it will have.

Author Response

We thank the reviewer for the comments and opinions.  We hope that eventually a newer generation of numerically savvy researchers will find our methods not so cumbersome.  It is our feeling that with use and practice, these methods are only a small step removed from earlier generation polynomial fitting of data.

Reviewer 4 Report

I am unable to add my report her. I have attached the file below and am sending it via email.

Author Response

We appreciate the reviewers attention to the details in the manuscript

"numerous" minor edits (too many to enumerate here) have been undertaken to improve the flow of the text.  Caption 3 has been revised to more clearly state what a trade off curve is - a concept that is less familiar in the mineral physics community than in the wider geophysics community We have undertaken a "meta-analysis" - combining many measurements. Since all measurements have errors, it is not useful here to undertake a detailed "forensic" analysis of what the errors are in the individual studies (other that two studies we removed for reasons that are well known in the community and are clearly stated here). Rather we are seeking to see what the aggregated data tell us about the properties office VII-X. As a result of our analysis, we believe that "state of the art" uncertainties in the ice VII-X data, extending to over 100 GPa are better than 5% in pressure.  It will be up to the next generation of experimentalists to determine whether they can beat that uncertainty down.  Our suggested mechanisms for the hypothetical rate-dependent process are stated starting on line 486 as well as in the abstract and conclusion. DFT results figure into multiple lines of investigation.  They provide P-V data as shown in Figure 2. They show up in Table 1 and discussions where mechanisms of transition and stability limits are discussed. Note that the revised manuscript adds several papers (French et al, Millot et al, and Myint et al) that were inadvertently dropped in the original submission.